# Experimental Production of Iron-Bearing Sinters Using Chars from Waste Car Tires

**Marian Niesler [1], Janusz Stecko [1], Damian Gierad [2], Martyna Nowak [3] and Sławomir Stelmach [3,\*]**

[1] Łukasiewicz Research Network—Institute for Ferrous Metallurgy, 44-100 Gliwice, Poland
[2] Energoekspert Sp. z.o.o., 40-145 Katowice, Poland
[3] Institute of Energy and Fuel Processing Technology, 41-803 Zabrze, Poland
[\*] Correspondence: sstelmach@itpe.pl

**Abstract:** The metallurgical industry is seeking raw material substitutes more and more intensively in order to replace materials traditionally used in pig iron production. Research has been conducted on the use of char obtained from waste car tires via a pyrolysis process in an iron ore sintering process. The char obtained from car tires could be a potential substitute for some of the coke breeze used in the iron ore sintering process. However, the Zn and S content of the char is a major technological issue. This paper presents the results of research conducted to assess the possibility of substituting coke breeze with a commercial char from waste tires. The experiments were carried out in a laboratory stand capable of sintering 200 kg of sintering blend. The results obtained show that it is possible to replace 10 %m/m of coke breeze with waste tire char without any technological danger for sintering lines. The application of waste tire char in metallurgical processes is an example of actions that form part of the circular economy and also of the appropriate use of anthropogenic resources that are technologically available.

**Keywords:** waste car tires; pyrolysis; char; coke breeze; iron-bearing sinters

## 1. Introduction

In developed economies, the automotive industry generates increasing quantities of waste tires, which are considered very difficult to biodegrade. Waste car tires account for 80 %m/m of all collected rubber waste [1]. The construction of car tires is complex, and a number of materials, mainly rubber, are used in their production. The rubber accounts for about 40–50 %m/m of the morphological composition, followed by soot (20–25 %m/m), relatively small amounts of construction elements (steel, fibers, etc.), and other additives, such as stabilizers and antioxidants [2–4]. The main vulcanizing substance used in the production of car tires is sulfur, which, together with zinc oxide and fatty acids, allows a vulcanization process to control and upgrade the physical properties of the rubber [5,6]. The sulfur and zinc oxide content in passenger car tires amounts to 1 %m/m; however, in the case of truck tires, the zinc oxide content is about 2 %m/m [2,3]. Waste car tires have a high carbon and hydrogen content, and hence their calorific value is very high at about 31–32 MJ/kg. Such high energy parameters make waste car tires very attractive energy carriers. Table 1 presents a comparison of the calorific value of car tires with other combustible materials.

There are numerous approaches to waste tire management, including retreading, material recycling, and energy recovery. However, waste tire management remains a problem because of the high amounts of waste generated annually and the need for the implementation of new methods of rational waste management [1,3]. It should not be forgotten that the thermal conversion of waste tires can cause the emission of hazardous compounds into the atmosphere, and such processes should always be carried out with the utmost care in order to protect the natural environment [7].

**Table 1.** Comparison of calorific values of selected combustible materials [1,3,8–10].

| Combustible Material | Calorific Value (MJ/kg) |
|---|---|
| Biomass | 15.1 |
| Paper/cardboard | 17.4 |
| Textiles | 18.4 |
| Hard coal | 26.4 |
| Anthracite | 27.8 |
| Coke breeze | 28.5 |
| Coke | 29.5 |
| Waste tires | 31.4 |
| Crude oil | 39.5 |

The metallurgical industry is increasingly seeking substitutes for commonly used raw materials. To meet the needs of the metallurgical industry, research on the possibility of using char from waste tires in the iron ore sintering process was carried out. The sintering process is commonly used in integrated steel plants to recycle carbon-containing residues using a blast furnace [11]. Over a billion tons of pig iron is generated annually in blast furnaces, and the basic raw material containing the iron is iron-bearing sinter. For this reason, it is crucial to seek technical and environmental optimization of the process. New fuels, which can be substituted for coke breeze in the iron ore sintering process, are being constantly sought [12–14]. At present, the alternative to coke breeze is anthracite, the consumption of which represents about 20–30 %m/m of the total mass of fuel used in the iron ore sintering process. The use of anthracite has a beneficial effect on the efficiency and economy of the sintering process, and the environmental impact is neutral [15–17].

The iron ore sintering tests described in this article enable assessment of the impact of char from waste car tires on the resulting sinter quality while maintaining the appropriate process effectivity. Waste car tires are thermally converted in the pyrolysis process that produces the char. The pyrolysis of waste car tires is a well-known process that has been described in many publications [3,5,8,18,19]. Generally, during the pyrolysis process, the following products are generated: ~38–55 %m/m of oil fraction, ~33–38 %m/m of char, and ~10–30 %m/m of a gaseous fraction [6,18,20]. After the appropriate processing, the pyrolytic oil (as a result of its high sulfur content) can be a liquid fuel because its calorific value reaches over 40 MJ/kg. The pyrolytic oil is a blend of organic compounds C6-C24, including benzene, toluene, xylenes, limonene, and derivatives of naphthalene, phenanthrene, fluorine, or diphenyl [5,8]. Pyrolytic gas consists mainly of hydrogen, light hydrocarbons C1-C4, carbon oxide, and carbon dioxide. Its calorific value is about 35 MJ/Nm³, and it is most often used as an energy carrier in the pyrolysis process [6,8]. The char from waste tires is the most difficult pyrolysis product to manage because of its sulfur and zinc oxide content [20]. Char is a light and brittle material consisting mainly of carbonized organic fractions with carbon content in the range of ~80–85 %m/m [6,21]. Its physicochemical properties are similar to those of technical soot [3], and its calorific value is high, reaching about 30–32 MJ/kg [21,22]. Many tests regarding the rational management of char have been conducted, but no industrial-scale management method has been developed. However, there is potential for an industrial application of char from waste tires as a substitute for coke breeze used as a fuel in the preparation of iron-bearing sinters for pig iron smelting in a blast furnace.

In Poland, there are several companies that perform the pyrolysis of waste car tires on a commercial basis. Each of these companies converts ~10–40 Mg of waste tires daily [23]. It is estimated that the annual production of tire char in Poland reaches ~20–25 thousand Mg, and in many cases, there are serious problems associated with its further management. Only one company, Reoil Sp. z o.o., generates char, which is then used as a substitute for soot produced by traditional methods [24]. The problem of accumulated

char from waste tires in Poland became a premise for experiments in the field of their management as a coke breeze substitute in the iron ore sintering process. The results of the experiments are presented below. The pyrolysis of waste car tires can be considered an element of the circular economy.

## 2. Materials and Methods

Table 2 provides a chemical analysis of coke breeze (CB) typically used in the iron ore sintering process and two samples of char (TC1 and TC2) from waste car tires received from Polish producers. The table shows only the parameters that are crucial from the sintering process point of view.

**Table 2.** Chemical analysis of tested fuels.

| Parameter | Coke Breeze (CB) %m/m | Tire Char (TC1) %m/m | Tire Char (TC2) %m/m |
|---|---|---|---|
| $Al_2O_3$ [1] | 3.36 | 0.43 | 1.20 |
| CaO | 1.55 | 1.95 | 1.30 |
| Cd | <0.01 | <0.01 | <0.01 |
| Co | <0.01 | <0.01 | <0.01 |
| Cu | <0.01 | <0.005 | 0.02 |
| Fe | 1.73 | 0.36 | 0.39 |
| K | 0.039 | 0.16 | 0.09 |
| MgO | 0.48 | 0.065 | <0.01 |
| Na | 0.12 | 0.06 | 0.13 |
| Ni | <0.01 | <0.01 | 0.02 |
| Pb | <0.01 | 0.020 | 0.002 |
| $SiO_2$ | 6.24 | 2.76 | 14.10 |
| Zn | <0.01 | 1.93 | 3.09 |
| C [2] | 81.0 | 76.00 | 74.50 |
| S | 0.9 | 2.27 | 2.69 |
| Cl [3] | 0.083 | 0.40 | 0.08 |
| Hg (ppm) | n.t. | <0.1 | <0.1 |
| Oil [4] | <0.01 | 16.5 | 0.45 |

[1] OES-ICP—Inductively coupled plasma optical emission spectrometry used for $Al_2O_3$, CaO, Cd, Co, Cu, Fe, K, MgO, Na, Ni, Pb, $SiO_2$, and Zn determination. [2] Coulometry method for C and S determination. [3] Spectrophotometric method for Cl determination, mercury analyzer for Hg. [4] Waste oil from pyrolysis process, weight method.

Both chars from waste tires contained a similar amount of elemental carbon, which has a direct impact on the calorific value and is a crucial parameter in the iron ore sintering process in terms of use as a coke breeze substitute. The sulfur and zinc content are significant parameters from the environmental and failure-free operation point of view and determines the use of the char in the iron ore sintering process. The sulfur content in the tested samples was 2.27 and 2.69 %m/m, respectively, whereas the zinc content was 1.93 and 3.09 %m/m. Another significant parameter for the sintering process is the pyrolytic oil content of the char samples because its presence is undesirable. During the process, part of the oil is combusted in the sintering blend bed, and the rest is carried out in the form of vapors with exhaust gases. As the temperature of the exhaust gases decreases, the oil vapors settle on dust particles, pipelines, and dedusting devices. Such phenomena can cause the deterioration of pipelines and other devices and, in the case of sinter plants using electrostatic precipitators, can cause the dust that has settled on the construction elements of the precipitators to catch fire [25].

The experiments were conducted on a semi-industrial line to simulate the sintering process. The line was equipped with an innovative exhaust gas neutralization system, which belonged to Łukasiewicz Research Network, Institute for Ferrous Metallurgy, Gliwice (Poland). The scheme of the research installation is presented in Figure 1, and a photograph of the semi-industrial-scale installation for sintering iron ores is presented in Figure 2.

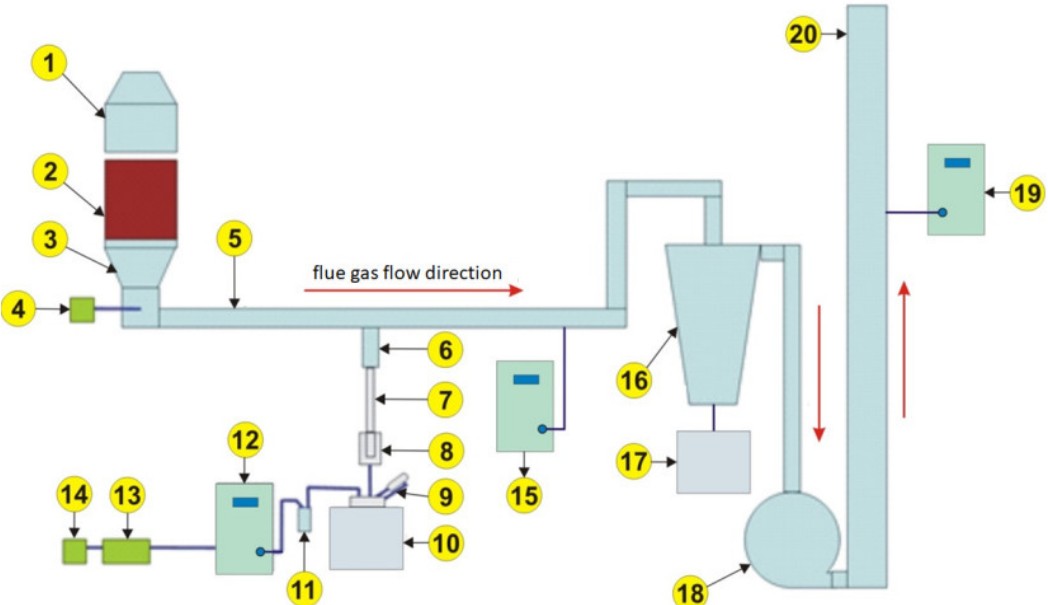

**Figure 1.** Scheme of the semi-industrial-scale installation for sintering of iron ores.

1. Natural gas burner
2. Sintering bed
3. Temperature measurement
4. Dust collection before the ceramic filter input
5. Flue gas pipeline
6. Collection point (stub pipe)
7. Thermostated probe
8. Cellulose filter (filter cup)
9. Absorber of organic compounds
10. Condenser with bottle for condensate

11. Moisture absorber
12. Suction pump with control system
13. Exhaust gases analyzer GA40Tplus
14. Exhaust gases analyzer GA20
15. Analyzer—continuous raw gas analysis
16. Ceramic filter
17. Dust collection from ceramic filter tank
18. Fan
19. Analyzer—continuous cleaned gas analysis
20. Stack

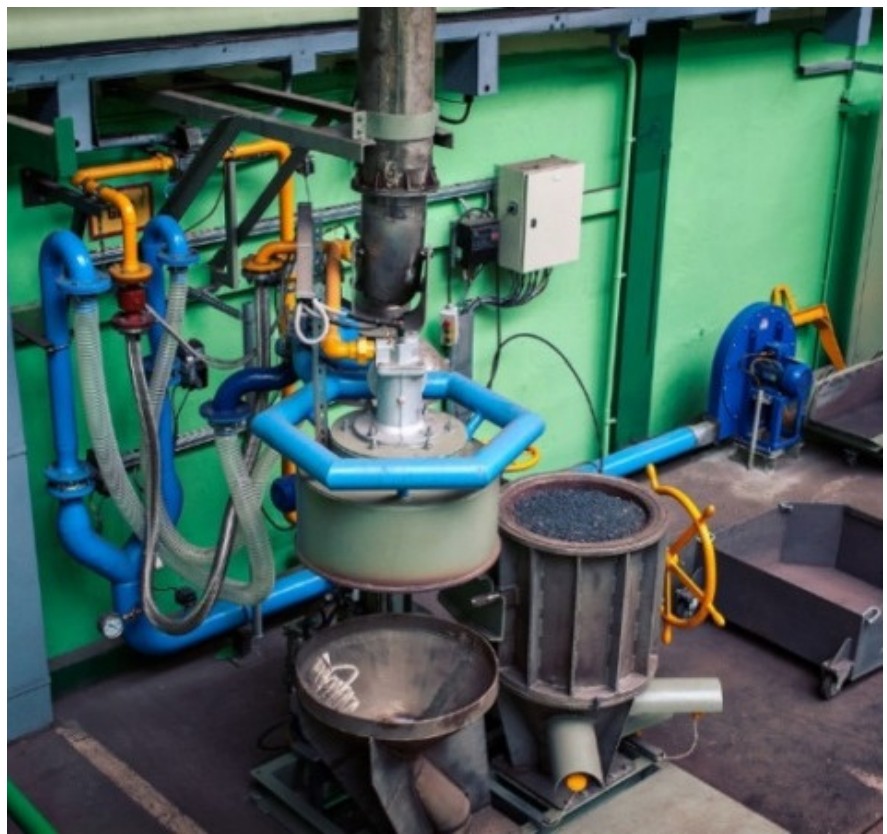

**Figure 2.** View of semi-industrial-scale installation for sintering of iron ores.

Sintering process tests were carried out using all the procedures and conditions that are applied for industrial sintering belts operated at integrated steel plants, i.e., the composition of the sintering mixture, retention time in the ignition furnace, pressure, sintering mixture height, addition of calcium oxide, mixture basicity, and amount of sinter return. The prepared blends for the sintering process contained iron ores in the form of concentrates, low silica ores and fine-grained iron ore, fluxes (limestone and dolomite), and coke breeze or its blends with the char from car tires. A sintering mixture (BM) with the following parameters was used: the ratio of hematite ore to magnetite concentrate, 0.82; basicity ($CaO/SiO_2$), 1.2; and magnesium oxide (MgO) content, 1.3 %m/m.

Table 3 shows the chemical composition of particular noncombustible components of the sintering blend. Table 4 presents examples of the composition of the formed sintering blends in order to compare the sinter without char (BM1) and the sinters with 10 and 20 %m/m char (TC1) of the total fuel mass (BM1.10 and BM1.20).

**Table 3.** Chemical composition of noncombustible components of the sintering blend.

| Chemical Component | Unit | Component of Sintering Blend | | | | |
|---|---|---|---|---|---|---|
| | | Krivbas Ore | KR Concentrate | Quicklime | Limestone | Dolomite |
| Fe | %m/m | 61.65 | 65.72 | 0.63 | 1.25 | 2.21 |
| $Fe^{2+}$ | %m/m | 0.57 | 27.00 | 0.00 | 0.50 | 0.80 |
| $SiO_2$ | %m/m | 9.97 | 8.00 | 0.50 | 0.94 | 0.86 |
| CaO | %m/m | 0.081 | 0.140 | 95.40 | 53.92 | 31.60 |
| $Al_2O_3$ | %m/m | 0.68 | 0.17 | 0.07 | 0.34 | 0.19 |
| MgO | %m/m | 0.120 | 0.351 | 0.300 | 0.590 | 20.34 |
| S | %m/m | 0.012 | 0.035 | 0.000 | 0.149 | 0.030 |
| $K_2O$ | %m/m | 0.013 | 0.021 | 0.000 | 0.000 | 0.000 |

| | | | | | |
|---|---|---|---|---|---|
| Na$_2$O | %m/m | 0.210 | 0.027 | 0.000 | 0.000 | 0.000 |
| Cl | %m/m | 0.240 | 0.014 | 0.000 | 0.013 | <0.010 |
| Zn | %m/m | 0.005–0.006 | 0.003 | 0.020 | 0.010 | 0.100 |

**Table 4.** Composition of sintering blends with and without TC1 char.

| Parameter | Unit | Comparative Blend (BM1) | TC1 Contribution in the Fuel, %m/m | |
|---|---|---|---|---|
| | | | 10 (BM1.10) | 20 (BM1.20) |
| KRIVBAS 59% | kg (wet) | 47.39 | 47.39 | 47.39 |
| KR concentrate | kg (wet) | 60.88 | 60.88 | 60.88 |
| Limestone | kg (wet) | 15.98 | 15.98 | 15.98 |
| Dolomite | kg (wet) | 5.88 | 5.88 | 5.88 |
| Quicklime | kg (wet) | 1.87 | 1.87 | 1.87 |
| Sinter return | kg (dry) | 56.2 | 56.2 | 56.2 |
| Solid fuel mass—sum (CB + TC1) | kg (wet) | 6.95 | 6.95 | 6.96 |
| Solid fuel mass—sum (CB + TC1) | kg (dry) | 6.13 | 6.20 | 6.27 |
| Solid fuel mass—CB | kg (dry) | 6.13 | 5.52 | 4.90 |
| Solid fuel mass—TC1 | kg (dry) | 0.00 | 0.69 | 1.37 |
| Fuel contribution in the blend | %m/m | 4.91 | 4.91 | 4.91 |
| Blend mass in the bed | kg (wet) | 186.30 | 185.40 | 186.07 |
| Moisture content in the blend | %m/m | 6.92 | 6.90 | 6.91 |
| Blend permeability (flow resistance) | s | 5.30 | 5.20 | 5.27 |

Sintering tests were conducted for the blend containing only coke breeze (the basic blend) and for the blends in which part of the coke breeze was replaced with car tire char (10 and 20 %m/m, respectively). It was assumed that the char contribution would be a maximum of 20 %m/m because of the undesirable zinc content. According to technological guidelines for blast furnace processes in Polish steel plants, the Zn content in the sinter should be lower than 0.015–0.020 %m/m.

## 3. Results and Discussion

Tables 5 and 6 set out the technological parameters of the sintering process and the quality of the sinters obtained from the experiments. BM1 was a comparative sinter without the addition of char. BM1.10 and BM1.20 were sintered with the addition of 10 %m/m and 20 %m/m of TC1 char, respectively. BM2 was a comparative sinter without char addition, and BM2.10 and BM2.20 were sintered with the addition of 10 %m/m and 20 %m/m of TC2 char, respectively.

**Table 5.** Technological parameters of the sintering process for BM1, BM1.10, and BM1.20 and sinter quality.

| Parameter | Unit | BM1 | BM1.10 | BM1.20 |
|---|---|---|---|---|
| | | Products | | |
| | | BS1 | S1.10 | S1.20 |
| *Sintering process parameters* | | | | |
| Sintering time | min | 21.92 | 22.27 | 22.10 |
| Production efficiency | Mg/m$^2$/24 h | 37.06 | 36.23 | 36.72 |
| Unit consumption of coke breeze | kg/Mg of sinter | 57.6 | 52.2 | 46.1 |

| Parameter | Unit | BM1 | BM1.10 | BM1.20 |
|---|---|---|---|---|
| | | Products | | |
| | | BS1 | S1.10 | S1.20 |
| *Sintering process parameters* | | | | |
| Unit consumption of char | kg/Mg of sinter | 0.0 | 6.5 | 12.9 |
| Total fuel consumption | kg/Mg of sinter | 57.6 | 58.7 | 59.04 |
| Maximal exhaust gases temperature | °C | 350.5 | 364.6 | 361.4 |
| *Sinter mass* | | | | |
| Sinter mass (>5 mm) | kg | 107.4 | 106.6 | 105.5 |
| Sinter return | kg | 51.3 | 51.2 | 53.0 |
| Return: 3–5 mm | kg | 25.0 | 24.4 | 26.6 |
| Return: 1–3 mm | kg | 14.6 | 15.2 | 16.7 |
| Return: <1 mm | kg | 11.7 | 11.6 | 9.6 |
| Sinter sum (return + sinter > 5 mm) | kg | 158.7 | 157.7 | 158.5 |
| *Sinter screen analysis* | | | | |
| >40 mm | %m/m | 13.19 | 14.88 | 12.4 |
| >25 mm | %m/m | 22.29 | 22.43 | 25.6 |
| >15 mm | %m/m | 22.69 | 22.87 | 23.8 |
| >10 mm | %m/m | 18.31 | 16.89 | 16.7 |
| >5 mm | %m/m | 23.52 | 22.93 | 21.4 |
| Median | mm | 18.02 | 18.84 | 19.4 |
| *Sinter strength* | | | | |
| Strength ISO TI | %m/m | 70.81 | 71.89 | 72.1 |
| Abrasibility ISO AI | %m/m | 5.77 | 5.74 | 5.70 |
| Drop breakability | %m/m | 32.32 | 32.44 | 33.5 |
| Mechanical drop strength | %m/m | 81.81 | 81.78 | 80.7 |
| *Reduction degradation index (RDI)* | | | | |
| <3.15 mm | %m/m | 19.2 | 19.1 | 18.3 |
| *Reducibility index (RI)* | | | | |
| dR/dt(O/Fe = 0.9) | %/min | 1.01 | 1.01 | 1.13 |

**Table 6.** Technological parameters of the sintering process for BM2, BM2.10, and BM2.20 and sinter quality.

| Parameter | Unit | BM2 | BM2.10 | BM2.20 |
|---|---|---|---|---|
| | | Products | | |
| | | BS2 | S2.10 | S2.20 |
| *Sintering process parameters* | | | | |
| Sintering time | min | 21.78 | 22.51 | 21.44 |
| Production efficiency | Mg/m$^2$/24 h | 36.90 | 35.85 | 37.01 |
| Unit consumption of coke breeze | kg/Mg of sinter | 58.4 | 52.2 | 47.2 |
| Unit consumption of char | kg/Mg of sinter | 0.0 | 6.2 | 12.7 |
| Total fuel consumption | kg/Mg of sinter | 58.4 | 58.4 | 59.89 |
| Maximal exhaust gases temperature | °C | 357.8 | 357.1 | 363.7 |
| *Sinter mass* | | | | |
| Sinter mass (>5 mm) | kg | 106.2 | 105.5 | 102.8 |
| Sinter return | kg | 51.1 | 52.4 | 52.5 |
| Return: 3–5 mm | kg | 24.6 | 25.1 | 25.4 |
| Return: 1–3 mm | kg | 14.8 | 15.4 | 15.9 |
| Return: <1 mm | kg | 11.8 | 11.9 | 11.2 |
| Sinter sum (return + sinter > 5 mm) | kg | 157.3 | 157.9 | 155.3 |

| Parameter | Unit | BM2 | BM2.10 | BM2.20 |
|---|---|---|---|---|
| | | Products | | |
| | | BS2 | S2.10 | S2.20 |
| *Sintering process parameters* | | | | |
| *Sinter screen analysis* | | | | |
| >40 mm | %m/m | 10.32 | 12.32 | 12.26 |
| >25 mm | %m/m | 20.84 | 22.66 | 23.94 |
| >15 mm | %m/m | 23.83 | 23.78 | 23.03 |
| >10 mm | %m/m | 20.19 | 18.02 | 17.09 |
| >5 mm | %m/m | 24.82 | 23.23 | 23.68 |
| Median | mm | 16.68 | 18.13 | 18.41 |
| *Sinter strength* | | | | |
| Strength ISO TI | %m/m | 72.17 | 73.33 | 73.08 |
| Abrasibility ISO AI | %m/m | 5.23 | 5.28 | 5.28 |
| Drop breakability | %m/m | 32.46 | 33.19 | 33.81 |
| Mechanical drop strength | %m/m | 81.47 | 80.49 | 80.81 |
| *Reduction degradation index (RDI)* | | | | |
| <3.15 mm | %m/m | 13.9 | 13.7 | 13.0 |
| *Reducibility index (RI)* | | | | |
| dR/dt(O/Fe = 0.9) | %/min | 1.01 | 1.08 | 1.14 |

Analysis of the data presented in Tables 5 and 6 shows that the addition of char influences the sintering process efficiency (Figure 3). In the case of the 10 %m/m TC1 and TC2 contribution in the fuel, there was an efficiency decrease from 37.06 to 36.23 Mg/m$^2$/24 h (a decrease of 0.83 Mg/m$^2$/24 h) and from 36.9 to 35.85 Mg/m$^2$/24 h (a decrease of 1.05 Mg/m$^2$/24 h), respectively. In the case of the 20 %m/m addition of TC1, the decrease was smaller, only 0.34 Mg/m$^2$/24 h. However, the 20 %m/m addition of TC2 increased the efficiency by 0.11 Mg/m$^2$/24 h. Higher efficiency can be achieved by using input material with better permeability and hence better utilization of the heat from the char.

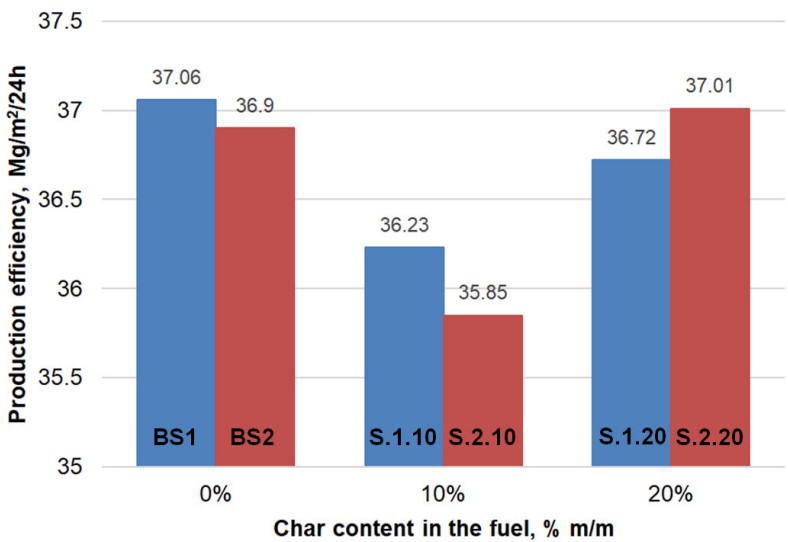

**Figure 3.** Changes in production efficiency depending on the content of char from waste tires in the fuel.

It is noticeable that the use of char as a coke breeze substitute slightly increased the unit fuel consumption: for the 10 %m/m and 20 %m/m addition of TC1, an increase of 1.1

kg/Mg and 1.44 kg/Mg of sinter, respectively. For the 10 %m/m addition of TC2, the fuel consumption remained at the same level, and for the 20 %m/m addition of TC2, the consumption increased by 1.49 kg/Mg of sinter (Figure 4).

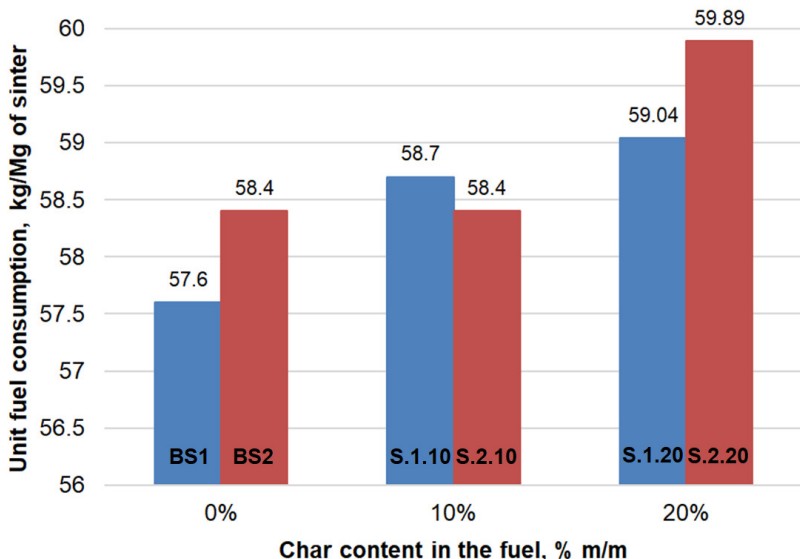

**Figure 4.** Unit fuel consumption depending on the content of char from waste tires in the fuel.

A sinter with a particle size greater than 5 mm is an input for the blast furnace process. In this research, the amount of this type of sinter was slightly lower when char was added than for the basic blend, as shown in Tables 5 and 6. For comparison, when char from biomass was added, the amount of particles > 5 mm was higher: 23,94–26,84 %m/m, depending on the biomass type [14]. For BM1 and BM2, the amount of sinter generated was 107.4 and 106.2 kg, respectively. In the case of a 10 %m/m addition of char, the mass of the generated sinter was 106.6 (for S1.10) and 105.5 kg (for S2.10). In the case of 20 %m/m addition of char, the sinter mass with particles > 5 mm was 105.5 (for S1.20) and 102.8 kg (for S2.20).

It should be noted that the sinter with added char had appropriate granularity and strength properties, as shown in Tables 5 and 6. Screen analysis of the sinters confirmed that those with added char had better granularity than the basic blends. The median particle sizes for sinters from the basic blends were 18.02 and 16.68 mm, whereas, in the case of a 20% addition of char, the medians were 19.4 and 18.41 mm.

The results obtained were also analyzed for strength and abrasibility properties using the tumble drum method, according to the ISO 3271:2015 standard [26], which determines the methods of iron ore strength assessment. The ISO TI (tumble index) and ISO AI (abrasion index) were determined. The ISO TI index was higher in sinters with added char in comparison with the basic blend. However, the S2.10 sinter was slightly stronger than the S2.20, which had a higher char content (20 %m/m) (Figure 5). The abrasibility of sinters generated from the blends with the addition of TC1 char slightly decreased, whereas, in the case of TC2, the abrasibility slightly increased in comparison to the basic blend. However, the abrasibility changes were so slight that their impact on sinter behavior should be minimal, and likewise in the case of the actual metallurgical process.

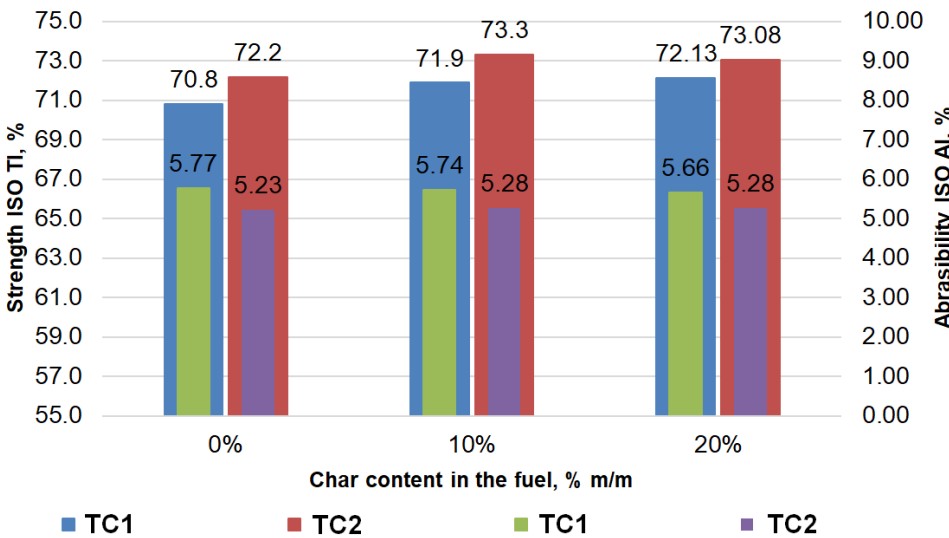

**Figure 5.** The impact of waste tire char on sinter strength (ISO TI) and abrasibility (ISO AI).

The reduction properties of sinters, as represented by RDI and RI indexes, are crucial from the blast furnace point of view. The Reduction Degradation Index (RDI) is an essential parameter used for sinter degradation prediction in the lower part of the blast furnace and is the reference for all sintering tests performed around the world. In order to perform the metallurgical process properly, the RDI should be as low as possible [27]. However, the reducibility index (RI) is a measure of oxygen transfer during the blast furnace process and delivers information on process fuel demand [27]. RDI results relating to the produced sinters showed that with the addition of char from waste tires, the amount of fine fraction <3.15 mm, generated in the blast furnace shaft at 500 °C, decreased as the char contribution in the fuel increased. This means that in this part of the blast furnace, the amount of fine sinter will be lower, which can result in a decrease in input permeability, and hence higher fuel consumption. The RDI for TC1 decreased from 19.2 %m/m for BM1 to 18.3 %m/m for S1.20. For TC2, the RDI decreased from 13.9 %m/m for BM2 to 13.0 %m/m for S2.20. The sinter reducibility index RI also improved after the addition of char to the fuel. The RI for the basic blends was 1.01%/min and increased to 1.13%/min for S1.20 and to 1.14%/min for S2.20. This means that in the blast furnace process, a sinter with higher reduction will reduce faster, and hence a smaller amount of fuel (reducer) will be needed. In comparison, when char from biomass is added, the RDI is 12.2–17.4%, depending on the biomass type [14].

According to blast furnace process guidelines in Polish steelmaking plants, the sinter should contain less than 0.015–0.020 %m/m of zinc. Tables 7 and 8 show average chemical analyses of sinter produced with char from waste tires.

**Table 7.** Average results of chemical analysis for sinter with TC1 in comparison to BS1 basic sinter.

| Parameter | Unit | BS1 | S1.10 | S1.20 |
|---|---|---|---|---|
| Fe | %m/m | 54.63 | 54.59 | 54.70 |
| $Fe^{2+}$ | %m/m | 8.13 | 7.60 | 6.60 |
| $SiO_2$ | %m/m | 9.253 | 9.253 | 9.22 |
| CaO | %m/m | 10.78 | 10.65 | 10.61 |
| Alkalinity ($CaO/SiO_2$) | - | 1.17 | 1.15 | 1.15 |
| $Al_2O_3$ | %m/m | 0.66 | 0.65 | 0.64 |
| $TiO_2$ | %m/m | 0.015 | 0.015 | 0.014 |
| MgO | %m/m | 1.38 | 1.31 | 1.29 |
| P | %m/m | 0.026 | 0.026 | 0.027 |
| Mn | %m/m | 0.024 | 0.025 | 0.024 |

| Parameter | Unit | BS1 | S1.10 | S1.20 |
|-----------|------|-----|-------|-------|
| S | %m/m | 0.017 | 0.019 | 0.024 |
| $K_2O$ | %m/m | 0.027 | 0.026 | 0.025 |
| $Na_2O$ | %m/m | 0.031 | 0.039 | 0.031 |
| Zn | %m/m | 0.012 | 0.023 | 0.037 |
| Cl | %m/m | 0.0117 | 0.0063 | 0.0083 |

**Table 8.** Average results of chemical analysis for sinter with TC2 in comparison to BS2 basic sinter.

| Parameter | Unit | BS2 | S2.10 | S2.20 |
|-----------|------|-----|-------|-------|
| Fe | %m/m | 53.85 | 53.90 | 53.84 |
| $Fe^{2+}$ | %m/m | 7.59 | 6.65 | 6.33 |
| $SiO_2$ | %m/m | 9.880 | 9.70 | 9.80 |
| CaO | %m/m | 11.45 | 11.47 | 11.44 |
| Alkalinity ($CaO/SiO_2$) | - | 1.16 | 1.18 | 1.17 |
| $Al_2O_3$ | %m/m | 0.54 | 0.53 | 0.52 |
| $TiO_2$ | %m/m | 0.012 | 0.012 | 0.013 |
| MgO | %m/m | 1.34 | 1.30 | 1.34 |
| P | %m/m | 0.019 | 0.020 | 0.019 |
| Mn | %m/m | 0.024 | 0.024 | 0.023 |
| S | %m/m | 0.025 | 0.029 | 0.032 |
| $K_2O$ | %m/m | 0.018 | 0.017 | 0.019 |
| $Na_2O$ | %m/m | 0.041 | 0.037 | 0.037 |
| Zn | %m/m | 0.011 | 0.028 | 0.049 |
| Cl | %m/m | 0.0132 | 0.0123 | 0.0130 |

Chemical analysis of the tested sinters showed that the addition of char from waste car tires does not strongly influence the basic parameters, i.e., Fe, $Fe^{2+}$ content, alkalinity, alkali, and chlorine content. Significant differences can, however, be observed in the sulfur and zinc content. In the sinter with added TC1, the sulfur content increased from 0.017 %m/m (BS1 sinter) to 0.024 %m/m (S1.20 sinter). In the case of zinc content, the increase was even more apparent. In BS1 and S1.12 sinter, the Zn content was 0.012 %m/m and 0.037 %m/m, respectively. The chemical analysis of TC2 char from waste tires showed that it contained more S and Zn than TC1, and this could be observed in the elemental analysis of the sinters. An increase in sulfur content from 0.025 %m/m for BS2 to 0.032 %m/m for S2.20 was observed. This tendency was much more noticeable with regard to the zinc content: BS2 and S2.20 sinter consisted of 0.011 %m/m and 0.049% of zinc, respectively.

In blast furnace conditions, the sulfur contained in the sinter must be removed into slag. Increased amounts of sulfur in sinter cause an increase in flux and limestone consumption and hence an increase in fuel consumption and $CO_2$ emissions to the environment resulting in increased blast furnace process performance costs.

Zinc content in processed inputs is not well tolerated in the blast furnace process, and for this reason, only a 10 %m/m addition of char from waste tires can be added to the fuel mass. Char contribution at this level does not exceed the permitted Zn content limit in the sinter, which is 0.015–0.020 %m/m. Of the tested chars, TC1 is better because it contains lower quantities of undesirable elements (Zn and S). The oil content of TC1 is high, but this can be reduced by the application of hydrated lime as a sorbent, whereas reducing the Zn is less straightforward. In comparison, Zn was lower in the case of the addition of char from biomass: 0.009–0.01%; however, its S content was 0.023–0.035%, depending on the type of biomass [14].

### 4. Conclusions

Our laboratory research conducted on iron ore and a waste sintering process simulation led to the following conclusions:

1. Char from waste tires applied as a partial coke breeze substitute should have high C content, but the Zn and S content should be low. It was assumed that because of the Zn content of TC1 (1.93 %m/m) and TC2 (2.27 %m/m), the contribution of char in the fuel should not exceed 20 %m/m.

2. A 20 %m/m contribution of TC1 in the fuel blend led to a decrease in production efficiency of 0.34 Mg/m$^2$/24 h in comparison with coke breeze. In the case of a 20 %m/m contribution of TC2, the production efficiency increased by 0.11 Mg/m$^2$/24 h, which could be because of the higher permeability of the input material (better use of the heat from the char).

3. A slight increase in unit fuel consumption was noted. In the case of a 10 %m/m contribution of TC1, consumption increased by 1.1 kg/Mg of sinter, and in the case of a 20 %m/m contribution, it increased by 1.44 kg/Mg. In the case of a 10 %m/m contribution of TC2, the fuel consumption remained at the same level, whereas a 20 %m/m contribution increased the consumption by 1.49 kg/Mg of sinter.

4. Sinters produced using char from waste tires had very suitable granularity and strength properties. ISO TI strength and ISO AI abrasibility were at the same level or slightly higher than for the basic blends.

5. The results of the chemical analyses of sinters produced using char were very similar to those of the basic blends, apart from the sulfur and zinc content. In the sinter with TC1, the sulfur content increased from 0.015 %m/m (for basic sinter BS1) to 0.024 %m/m (for sinter S1.20). In basic sinter BS1, the zinc content was 0.012 %m/m and increased to 0.037 %m/m for sinter S1.20. For TC2, the increase in the sulfur content was from 0.025 %m/m (for basic sinter BS2) to 0.032 %m/m (for sinter S2.20). The increase was even more noticeable in the case of the zinc content, with the basic sinter containing 0.011 %m/m Zn and the sinter with the addition of 20 %m/m of char TC2 containing 0.049 %m/m.

6. The high zinc content of sinters produced from blends of coke breeze and char means that only 10 %m/m of char from waste tires can be added to the fuel mass in order to ensure that the zinc content equals 0.015–0.020 %m/m. Of the tested chars, TC1 is better because the Zn and S content is lower, and the high oil content can be reduced by adding hydrated lime as a sorbent.

7. The experiments performed showed that char from waste car tires can actually be used in the process for the production of iron-bearing sinters. Even a 10 %m/m contribution of such combustible waste in the input fuel blend for the sintering process allows a huge amount of char from waste car tires to be managed, which at present is otherwise challenging.

**Author Contributions:** Conceptualization, M.N. (Marian Niesler), J.S., D.G., and S.S.; methodology, M.N. (Marian Niesler), J.S., and S.S.; investigation, J.S., D.G., and M.N. (Martyna Nowak); data curation, M.N. (Marian Niesler), J.S., D.G., and S.S.; writing—original draft preparation, M.N. (Marian Niesler), J.S., and D.G.; writing—review and editing, S.S. and M.N. (Martyna Nowak); visualization, S.S. and M.N. (Martyna Nowak); supervision, S.S. All authors have read and agreed to the published version of the manuscript.

**Funding:** This research received no external funding.

**Institutional Review Board Statement:** Not applicable.

**Informed Consent Statement:** Not applicable.

**Data Availability Statement:** Not applicable.

**Conflicts of Interest:** The authors declare no conflict of interest.

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
