# Peer review of "Experimental Production of Iron-Bearing Sinters Using Chars from Waste Car Tires"

_processes, doi:10.3390/pr11010231_

Round 1
Reviewer 1 Report
The topic is relevant and presents a high degree of novelty and interest. The procedure is complete and the performed analysis techniques acqurate.
1. Fig.1 is not clear, please revise it.
2. The conclusion is too much, please simplify the expression.
Author Response
Thank you very much for the generally positive evaluation of our publication. With regard to the comments presented, we would like to inform you that Figure 1 has been divided into two separate figures (Figure 1 and Figure 2) for clarity. Conclusions have also been slightly modified. We hope that the changes we have made are sufficient. The entire publication has been corrected in terms of English language using editing service offered by MDPI.
Reviewer 2 Report
This study investigated the performance of two types of waste-tire-derived char in the iron ore sintering process. The results provided new insights into utilizing waste tire pyrolysis char. However, the manuscript still needs to be revised.
Specific comments
1. The English level of the manuscript needs to be improved.
2. Literature citations are not standardized. The authors need to cite more literature other than their own.
3. Page 1 Lines 36–38: This sentence is hard to read. Please rewriting this sentence or splitting it into two sentences.
4. Page 1 Lines 40–41: Please revise "There is a numerous ways" into " There are numerous ways".
5. Page 2 Lines 48–49: Please revise this sentence into "The metallurgical industry is increasingly seeking substitutes for commonly used raw materials."
6. Page 2 Lines 84–86: What is the unit of Mg? Please use standard international (SI) units.
7. Page 3 Lines 91–93: This sentence is hard to read. Please rewriting this sentence.
8. Page 3 Lines 96–97: The authors investigated the performance of two types of char in the iron ore sintering process. Indeed, the char samples were obtained from commercial waste tire pyrolysis companies. It would be desirable to provide the operating conditions during char production, such as temperature, residence time, reactor types, etc.
9. Page 8 Lines 176–178: This sentence is hard to read. Please rewriting this sentence.
10. Page 8 Lines 183–184: There is no need to add the comma before "that".
11. The conclusions need to condense. Points 1 and 5 have too much content.
Author Response
Thank you very much for the generally positive evaluation of our publication. With regard to the comments presented, we would like to inform you that we changed the article according to your suggestions. The entire publication has been corrected in terms of English language using editing service offered by MDPI. Literature citations in the publication are made in accordance with the template file downloaded from the MDPI system.
The sentence in page 1 lines 36-38 is changed into: Waste car tires have high carbon and hydrogen content and hence the calorific value is very high – about 31-32 MJ/kg. Such high energy parameters considers waste car tires as very attractive energy carriers.
The sentences in lines 40-41 and 48-49 are revised as suggested.
The unit Mg is megagrams and is a derived standard unit of mass in SI unit standard equals to 1000 kilograms.
The sentence in lines 91-93 is rewritten into: The pyrolysis of waste car tires can be considered as an element of the circular economy.
Considering the sentence in lines 96-97 – unfortunately we do not received the operating conditions of the char production from the commercial pyrolysis companies. The installations from which the samples were obtained were installations operating in the periodic system. The maximum conversion temperature is in the range of ~550-600oC, and the residence time of the charge in the reactor is ~10-12h.
The sentence in lines 176-178 is rewritten into: The sinter with desired particles size higher than 5 mm is an input for the blast furnace process. In this research the amount of this type of sinter is slightly lower in case of char addition than for the basic blend, as shown in table 5 and 6.
The commas before “that” are deleted. Conclusions have been somewhat shortened.
We hope that the changes we have made are sufficient.
Reviewer 3 Report
In manuscript, the effect of addition of chars derived waste tires, as substitute coke breeze, on the iron ore sintering process have been studied. Two chars having different Zn, S and oil content were tested. The sintering process efficiency, unit fuel consumption, sinters strength ISO TI and ISO AI abrasibility have been comparatively presented for two chars and addition ratios. The manuscript is a useful technical report. Although authors have a study on the use of chars derived residual biomass as a substitutional fuel in the iron ore sintering process (https://doi.org/10.3390/en14133749), they didn't mention it in the manuscript. The manuscript would have scientific value if they had compared and discussed the results.
- Some Footnotes in Figure 1 cannot be fully read.
- Although the 10% addition of TC1 increased the unit fuel consumption, the fuel consumption remains on the same level in case of 10% addition of TC2. Why, please explain.
- In the Fig.2, 3 and 4, it is not appropriate to use the TC1 and TC2 labels for 0% char content.
-In conclusion, they concluded that TC1 is better, since it contains less undesirable elements. But TCI has very high oil content. Is oil desirable?
Author Response
Thank you very much for the generally positive evaluation of our publication. With regard to the comments presented, we would like to inform you that we changed the article according to your suggestions. The entire publication has been corrected in terms of English language using editing service offered by MDPI.
According to the manuscript considering application of biomass chars, comparison of few parameters were presented in the text.
Figure 1 has been divided into two separate figures (Figure 1 and Figure 2) for clarity.
10% addition of TC2 did not increase the fuel consumption. It can be caused by proper permeability of the input and good energy use, in case of this share of the char. Only exceeding the amount of char in the blend causes a significant increase of fuel consumption.
In Figures 3 and 4, notations have been added to make them fully understandable to the reader.
With regard to the oil content: TC1 is better since it contains less undesirable elements – it is related only to Zn and S content. Especially, it is important for the Zn content to be as low as possible, since it cannot be easily removed from the sinter. The oil content is also undesirable, however it can be reduced by adding hydrated lime as a sorbent. The explanation is included in the conclusions.
We hope that the changes we have made are sufficient.
Round 2
Reviewer 2 Report
The manuscript can be accepted in present form.